# Nanostructured Lipid Carriers Enriched Hydrogels for Skin Topical Administration of Quercetin and Omega-3 Fatty Acid

**DOI:** 10.3390/pharmaceutics15082078

**Published:** 2023-08-03

**Authors:** Marlene Lúcio, Nicole Giannino, Sérgio Barreira, José Catita, Hugo Gonçalves, Artur Ribeiro, Eduarda Fernandes, Isabel Carvalho, Hugo Pinho, Fátima Cerqueira, Marco Biondi, Carla M. Lopes

**Affiliations:** 1CF-UM-UP, Centro de Física das Universidades do Minho e Porto, Departamento de Física, Universidade do Minho, 4710-057 Braga, Portugal; eduardabfer@gmail.com; 2CBMA, Centro de Biologia Molecular e Ambiental, Departamento de Biologia, Universidade do Minho, 4710-057 Braga, Portugal; 3Instituto de Investigação, Inovação e Desenvolvimento (FP-I3ID), Biomedical and Health Sciences Research Unit (FP-BHS), Faculdade Ciências da Saúde, Universidade Fernando Pessoa, 4200-150 Porto, Portugal; 32316@ufp.edu.pt (N.G.); barreira@ufp.edu.pt (S.B.); jcatita@ufp.edu.pt (J.C.); 37118@ufp.edu.pt (H.P.); fatimaf@ufp.edu.pt (F.C.); 4Dipartimento di Farmacia, Università degli Studi di Napoli Federico II, Via Domenico Montesano 49, 80131 Napoli, Italy; mabiondi@unina.it; 5Paralab, SA, 4420-392 Valbom, Portugal; hugo.goncalves@paralab.pt; 6CEB, Centro de Engenharia Biológica, Universidade do Minho, 4710-057 Braga, Portugal; arturibeiro@ceb.uminho.pt (A.R.); isabel.carvalho@deb.uminho.pt (I.C.); 7LABBELS, Associate Laboratory, Braga/Guimarães, Portugal; 8LIBRO—Laboratório de Investigação em Biofilmes Rosário Oliveira, University of Minho, Campus of Gualtar, 4710-057 Braga, Portugal; 9Molecular Oncology and Viral Pathology Group, Research Center of IPO Porto (CI-IPOP)/RISE@CI-IPOP (Health Research Network), Portuguese Oncology Institute of Porto (IPO Porto)/Porto Comprehensive Cancer Center (Porto.CCC), 4200-072 Porto, Portugal; 10Associate Laboratory i4HB—Institute for Health and Bioeconomy, Faculty of Pharmacy, University of Porto, 4050-313 Porto, Portugal; 11UCIBIO—Applied Molecular Biosciences Unit, MEDTECH, Laboratory of Pharmaceutical Technology, Department of Drug Sciences, Faculty of Pharmacy, University of Porto, 4050-313 Porto, Portugal

**Keywords:** quercetin, omega-3 fatty acids, skin topical administration, nanostructured lipid carriers (NLCs), antioxidant, photoprotective, antimicrobial effect

## Abstract

Chronic skin exposure to external hostile agents (e.g., UV radiation, microorganisms, and oxidizing chemicals) may increase oxidative stress, causing skin damage and aging. Because of their well-known skincare and protective benefits, quercetin (Q) and omega-3 fatty acids (ω_3_) have attracted the attention of the dermocosmetic and pharmaceutical sectors. However, both bioactives have inherent properties that limit their efficient skin delivery. Therefore, nanostructured lipid carriers (NLCs) and enriched PFC^®^ hydrogels (HGs) have been developed as a dual-approach vehicle for Q and/or ω_3_ skin topical administration to improve bioactives’ stability and skin permeation. Two NLC formulations were prepared with the same lipid composition but differing in surfactant composition (NLC1—soy lecithin and poloxamer 407; NLC2—Tween^®^ 80 and dioctyl sodium sulfosuccinate (DOSS)), which have an impact on physicochemical properties and pharmaceutical and therapeutic performance. Despite both NLCs presenting high Q loading capacity, NLC2′s physicochemical properties make them more suitable for topical skin administration and ensure longer colloidal stability. Additionally, NLC2 demonstrated a more sustained Q release, indicating higher bioactive storage while improving permeability. The occlusive effect of NLCs-enriched HGs also has a positive impact on skin permeability. Q-loaded NLC2, with or without ω_3_, -enriched HGs demonstrated efficacy as antioxidant and photoprotective formulations as well as effective reduction in *S. aureus* growth, indicating that they constitute a promising approach for topical skin administration to prevent skin aging and other damaging cutaneous processes.

## 1. Introduction

The skin is regarded as our most visible and prominent organ, and it serves an important defense function against the harmful external environment. The skin’s complex, stratified structure acts as a barrier, limiting the entry of microorganisms while also protecting against free radical oxidants and other external pollutants [1,2,3]. Aside from its protective function, the skin also regulates body temperature, maintains internal homeostasis, and acts as a sensory organ. Despite its inherent protective antioxidative system, chronic exposure of the skin to damaging radiation (e.g., UV exposure) and other external environmental aggressive agents (e.g., microorganisms and oxidizing chemicals) may boost the skin’s oxidative stress and readily induce skin damage and aging [2,4,5].

Since the advent of gut microbiome research and its paramount importance in the emergence and evolution of some diseases [6], the skin microbiome, a largely symbiotic community of commensal skin microorganisms, has been regarded as the next frontier in preventive skin health, with pertinent links being established between the microbiome and cutaneous repair or skin aging [7,8]. Natural antioxidants applied topically to the skin are a promising strategy for reducing oxidative stress and improving the status of the skin microbiome while retaining the skin’s protective qualities against external hostile agents. Among these natural antioxidants, quercetin (Q) is recognized for its plentiful skin care benefits, including anti-aging, anti-bacterial, anti-fungal, and photoprotection effects [2,9,10,11,12,13,14,15]. The FDA has granted Q GRAS (Generally Recognized As Safe) status [16]. Despite its broad-spectrum effects on the skin, there are several obstacles to Q topical administration, namely its limited aqueous solubility and high lipophilicity, which hinder its skin permeation [17,18,19]. Moreover, Q is chemically unstable and very susceptible to oxidation [2]. Lipid-based nanosystems are considered the most suitable for topical administration due to their small size and composition, which imitate the natural barriers of the skin [3,20]. These nanosystems have the ability to increase epidermal permeation by fluidizing the lipid matrix of the *stratum corneum* without impairing the functions of this skin barrier [21]. In this regard, loading Q into lipid-based nanosystems (e.g., liposomes, micro/nanoemulsions, solid lipid nanoparticles—SLNs—and nanostructured lipid carriers—NLCs) has been explored to increase Q skin permeation and stability. NLCs were selected among the various lipid-based nanosystems supported by our team’s previous expertise in developing skin topical formulations for photoprotection [22] or natural antioxidant administration (resveratrol) [23]. The previously developed NLCs also contained omega-3 fatty acids (ω_3_), which served a dual purpose, namely as an oil component of the NLCs’ lipid matrix and as a bioactive with beneficial skin care effects (e.g., photoaging protection, fluidizing qualities that promote penetration, and occlusive action). In the present study, we extended our expertise in these formulations by designing novel NLC (NLC1 or NLC2— Figure 1)-enriched hydrogels (HGs) for improving Q and/or ω_3_ stability and skin topical administration. HGs provide additional benefits for skin topical administration due to their 3D networks of hydrophilic polymers that can swell and absorb a high amount of water or biological fluids, preventing dissolution, enhancing skin adhesion and hydration, and prolonging bioactives’ release. In light of our current knowledge, very few reported lipid-based nanosystems were developed for Q and ω_3_ co-administration [24,25,26,27], and just one research project produced NLCs with both components [27]. Furthermore, none of these formulations were designed to be applied topically to the skin. Moreover, Q-loaded HGs have been reported for topical application in skin disorders, but none of these studies have provided NLC (or other lipid-based nanosystems)-enriched HGs as a dual strategic vehicle for topical skin Q and/or ω_3_ administration.

The development and physicochemical characterization of NLCs, including size, surface charge, polydispersity index (PDI), colloidal stability, Q encapsulation efficiency (EE), drug loading (DL) capacity, and thermodynamic analysis, were carried out in the first part of this study. The second part of this study investigated the potential pharmaceutical and therapeutic performance of the developed NLC-enriched HGs, specifically by studying their in vitro Q release and permeation through a mimetic *stratum corneum* lipid membrane, rheological behavior, antioxidant effect by free radical scavenging and protection against lipid peroxidation, photoprotective effect, and antimicrobial effect.

## 2. Materials and Methods

### 2.1. Materials

Quercetin (≥95% HPLC) (Q), omega-3 fatty acids (fish oil from menhaden) (ω_3_), and dioctyl sodium sulfosuccinate (DOSS) were purchased from Sigma Aldrich Co. (St. Louis, MO, USA). Soy lecithin (glycerophospholipids), Tween^®^ 80, triethanolamine, and glycerin were acquired from Acofarma^®^ (Madrid, Spain). Citric acid and Sodium phosphate dibasic dihydrate were acquired from Sigma Aldrich Co. (St. Louis, MO, USA).

Precirol ATO^®^ 5 (glycerol distearate), Gelucire^®^ 50/13 (stearoyl polyoxyl-32 glycerides), and Labrasol^®^ (caprylocaproyl polyoxyl-8-glycerides) were donated from Gattefossé (Saint-Priest Cedex, France). Poloxamer 407 (also known as Pluronic^®^ F127) was supplied by BASF (Ludwigshafen am Rhein, Germany). Gelling PFC^®^ (Carbopol 2001) was purchased from Guinama S.L.U. (Valencia, Spain). Methanol (≥99.9%), acetonitrile (HPLC gradient grade), and glacial acetic acid (≥99.7%) were acquired from Fisher Scientific (Loughborough, UK). Lipid dipalmitoylphosphatidylcholine (DPPC) was obtained from Avanti Polar Lipids, Inc. (Instruchemie, Delfzijl, The Netherlands). Cholesterol (Chol) and mixed cellulose esters filters (0.65 μm pore size) were acquired from Merck Life Science (Algés, Portugal).

Antioxidant and TBARS assay kits of Cayman Chemical were bought from Bertin Bioreagent (Lisbon, Portugal).

Tryptic Soy Broth (TSB), Sabouraud Dextrose Broth (SDB), and Agar were acquired from Liofilchem (Roseto degli Abruzzi, Italy). Neubauer hemocytometers were bought from Paul Marienfeld GmbH & Co. KG (Lauda-Königshofen, Germany).

All the reagents used were of analytical grade or the highest grade available.

### 2.2. Methods

#### 2.2.1. Preparation of Nanostructured Lipid Carriers

Given the vast array of lipids available to produce NLCs, the most promising core compositions were selected by the microscopical and macroscopical assessment of the physical compatibility between liquid and solid lipids, which was performed in a similar way to the lipid compatibility and miscibility procedures described by Basso et al. [28]. Q solubilization within the most promising lipids or lipid mixture at different wt% was also microscopically and macroscopically evaluated. The results for different lipids and lipid mixtures with 0.2% of Q are presented in Appendix A. The final NLC composition (NLC1 and NLC2—Appendix A) showed no macroscopic or microscopic signs of lipid compatibility, miscibility, or Q solubilization issues.

NLCs were prepared using a modified melt emulsification method followed by ultrasonication, as described by Mendes et al. [29]. The lipid phase (Appendix A) was heated in a water bath at a temperature 5–10 °C above the melting point of the solid lipids’ mixture for 20 min to guarantee the complete solubilization of Q in the lipid phase. The aqueous phase (Appendix A), heated at the same temperature, was added to the lipid phase, and homogenized under high-speed stirring (9000 rpm for 5 min) using an Ultra-Turrax^®^ T25 (IKA^®^, Janke & Kunkel GmbH, Staufen, Germany), followed by sonication using a probe sonicator (Bandelin Electronic UW 2200, Berlin, Germany) at 40% amplitude for 10 min. The resulting O/W nanoemulsion was immediately transferred to glass vials and cooled to room temperature in an ice bath to produce the NLCs. Each of the NLC1 and NLC2 formulations (Figure 1) was prepared in triplicate (three independent batches). Therefore, each analysis was performed in triplicate for each batch.

#### 2.2.2. Characterization of Nanostructured Lipid Carriers

##### Measurement of Particle Size, Polydispersity Index, and Zeta Potential

Dynamic and electrophoretic light scattering (DLS and ELS) were used to characterize NLCs in terms of particle size (Z-average), polydispersity index (PDI), and zeta potential using a Zetasizer Nano ZS (Malvern^®^, Worcestershire, UK). To avoid the multiple scattering effect caused by a high particle concentration, NLCs were diluted with purified water (*v*:*v*) (1:400 and 1:200 for NLC1 and 1:200 and 1:25 for NLC2), obtaining the appropriate scattering intensity (i.e., count rate of 250–500 and minimal attenuation). On the preparation day (T1), all experiments were carried out at a controlled temperature of 25 ± 1 °C. After cumulant analysis, size and PDI were calculated from the correlogram using the software Zetasizer Nano ZS (Version 8.02, Malvern^®^, Worcestershire, UK) and ISO 22412:2008 [30]. The zeta potential was calculated from electrophoretic mobility using the Helmholtz-von Smoluchowski method [31].

The above-mentioned characterization was also performed for four weeks (from T1 to T4) to evaluate the colloidal stability of NLCs under storage conditions (at 4 °C).

##### Differential Scanning Calorimetry Analysis

Differential scanning calorimetry (DSC) was performed using a NEXTADSC 600^®^ (HITACHI, Ibaraki, Japan) equipped with an automatic sample changer to investigate the physicochemical compatibility of the NLC components and the melting and crystallization behavior of the lipid dispersions. Bulk materials and mixtures of lipids and Q weighing between 8 and 12 mg were put into sealed aluminum crucibles with a perforated cap. As a reference, an equivalent empty crucible and cap were used. The thermal analysis profiles were obtained under a dynamic atmosphere of nitrogen (20 mL/min). For bulk materials and lipid mixtures with Q, the thermal program contemplated a cooling down to 0 °C (at a rate of 30 °C/min), followed by an isotherm of 5 min, and subsequently heating from 0 to 200 °C (at a rate of 5 °C/min). Data were obtained using the NEXTA Standard Analysis (version 2.7) software.

##### Encapsulating Efficiency and Loading Capacity

The encapsulating efficiency (EE) and drug loading (DL) capacity were determined indirectly by calculating the amount of non-encapsulated Q present in the aqueous phase of dispersions and applying the following equations:(1)$$EE(\%)=\frac{{\left[Q\right]}_{Total}-{\left[Q\right]}_{Free}}{{\left[Q\right]}_{Total}}\times 100$$
(2)$$DL(\%)=\frac{{\left[Q\right]}_{Total}-{\left[Q\right]}_{Free}}{{\left[Lipid\right]}_{Total}}\times 100$$

A total of 500 μL of Q-loaded NLCs (i.e., NLC1 + Q, NLC1 + Q + ω_3_, NLC2 + Q, and NLC2 + Q + ω_3_) were transferred to Ultracel 100 K centrifugal filter devices (Amicon^®^ Ultra, Millipore Corporation, Bedford, MA, USA). These filter devices were centrifuged (Labofuge 400 centrifuge, Thermo Scientific Heraeus^®^, Cacém, Portugal) at a rcf of 1808× *g* for 30 min. The amount of [Q]_Free_ was determined by a high performance liquid chromatography (HPLC) method that was developed based on the methods of Ang et al. [32] and Vijayakumar et al. [33]. An HPLC (HP Agilent 1100 HPLC System, Agilent Technologies (Waldbronn, Germany)) equipped with an UV–VIS detector and using an imChem (Voisins le Bretonneux, France) Surf C18 column (5 μm particle size; 150 × 4.6 mm i.d.) was employed. The chromatography was carried out at a flow rate of 1 mL/min under isocratic elution (mobile phase consisting of acetonitrile and 2% *v*/*v* acetic acid (pH 2.60) (40%:60% *v*/*v*)). All analyses were performed at a 370 nm wavelength and with an injection volume of 20 μL.

#### 2.2.3. Preparation of Nanostructured Lipid Carriers Enriched Hydrogels

The inclusion of NLCs in semisolid forms, such as HGs, appears to be a successful method for their topical administration given the low viscosity of lipid nanoparticles in aqueous dispersion [34]. NLCs were directly dispersed in the gelling agent, i.e., PFC^®^ 0.5% (*w/w*), in a porcelain mortar, and afterwards was neutralized with triethanolamine until the hydrogel was formed (≈pH 6.5). A humectant agent (glycerin 2.5% *w/w*) was added before equilibrating the hydrogel for 24 h in the refrigerator (4 °C).

#### 2.2.4. Evaluation of Pharmaceutical and Therapeutic Performance

##### In Vitro Drug Release Studies

Using a dialysis membrane (Float-A-Lyzer^®^, 3.5 kD, VWR) diffusion method, an in vitro Q release profile from NLCs was carried out. The release medium, which contains 65% buffer pH 5.6 (0.1 M citric acid and 0.2 M sodium phosphate dibasic dihydrate) and 35% absolute ethanol, was selected in order to ensure both the dissolution of Q and mimic the pH of the skin [35]. Approximately 1.0 g of Q-loaded NLC (i.e., NLC1 + Q, NLC1 + Q + ω_3_, NLC2 + Q, and NLC2 + Q + ω_3_)-enriched hydrogels (HGs) were weighted into dialysis membranes and immersed in 20 mL of dissolution medium, ensuring sink conditions. The system was maintained at the skin surface’s mimetic temperature of 32 ± 3 °C [36], using a SW22 Shaking water bath (Julabo GmbH, Seelbach, Germany) and stirring at a speed of 100 rpm. At scheduled time points, 1.0 mL aliquots of the receptor medium were withdrawn and replaced with fresh release medium at predetermined intervals until 24 h. These aliquots were diluted with methanol (1:1 dilution), and Q was quantified by the previously described validated HPLC method. The cumulative Q released was calculated and expressed as a percentage of the theoretical maximum Q content value.

##### In Vitro Permeability Studies

Q in vitro diffusion from the Q-loaded NLC-enriched HGs was evaluated using Franz diffusion cells (V-Series Stirrers for Franz Cells; PermeGear, Hellertown, PA, USA). Lipids are abundant in the interlamellar regions of the *stratum corneum* and act as a significant barrier to drug permeation [3]. Therefore, a mimetic *stratum corneum* lipid membrane was prepared in house over a commercial cellulose filter scaffold (Appendix A represents a morphological electron microscopy characterization of this model). The rationale for using this mimetic lipid matrix was previously described in papers that produced similar *stratum corneum* models (e.g., [37]). Additionally, the method used to deposit the DPPC:CHOL (2:1) mimetic model in a cellulose scaffold was developed via adaptation of the phospholipid vesicle-based permeation assay (PVPA), which is a simple and reproducible method for studying the fundamental mechanism of drug permeation through several physiological barriers, originally described by Flaten et al. [38]. Briefly, a lipid mixture composed of DPPC:Chol (2:1) was first dissolved in chloroform and then evaporated under a stream of nitrogen, resulting in a thin lipid film. The dried lipid film was hydrated in a water/ethanol (10% *v*/*v*) mixture, and the resultant suspension was submitted to 5 cycles of vortexing and heated at 60 °C to form multilamellar vesicles. Subsequently, multilamellar vesicles were deposited on mixed cellulose filters (0.65 μm pore size) by spin coating at 0.3 speed, followed by freeze-thaw cycles.

The lipid-coated membrane with a diffusion area of 0.64 cm^2^ was mounted in the Franz diffusion cells, between the donor and the receptor compartment (containing 5 mL of 65% buffer pH 5.6 and 35% absolute ethanol). The membrane was acclimatized at 32 ± 3 °C for 0.5 h prior to the addition of Q-loaded NLC-(i.e., NLC1 + Q, NLC1 + Q + ω_3_, NLC2 + Q, and NLC2 + Q + ω_3_) enriched HGs to the donor compartment, and the temperature was maintained constant throughout the assay using a circulating water bath. Receptor medium was continuously stirred for 12 h, and aliquots were collected at 1 h time intervals and replaced with the same volume of fresh medium.

Q concentration in the receptor compartment was determined using the validated HPLC method described in “Encapsulating Efficiency and Loading Capacity”.

The cumulative quantity of Q release per unit membrane area (*Q_R_/A*) versus time (*t*) plot was used to evaluate permeation parameters. The steady-state maximum flux of permeation (*J_ss_*) and lag time (*t_L_*) correspond, respectively, to the gradient and x intercept of the linear portion of the plot. *J_ss_* (μg cm^−2^ h^−1^) was determined using Equation (3) [39]:(3)$$J_{SS}=\frac{Q_{R}}{A\times t}$$
where *t* (h) is the permeation time, *A* (cm^2^) the permeation area, and *Q_R_* (μg) the permeated amount of drug.

##### Occlusive Effect Assessment

The in vitro occlusive effect was used to investigate the prevention of water loss by NLC-enriched HGs [40]. A closed glass container with an accurate mass of deionized water was covered with a Whatman^®^ cellulose microfiber filter (pore size from 0.6 to 0.8 μm) and sealed with parafilm. Equivalent amounts of NLC-enriched HGs were evenly distributed over the filter surface. The systems were kept in an incubator for 24 h at 32 ± 3 °C. The system was accurately weighed at predetermined time intervals to find out the weight loss of water, i.e., the water vapor transmission rate (WVTR) expressed in g·m^−2^·day^−1^ [41]:(4)$$WVTR=\frac{w_{i}-w_{f}}{t\times A}$$
where *t* represents the 24 h time point, *A* is the sample testing area (m^2^), and *w_i_* and *w_f_* are the initial and final weight of the system, respectively.

The occlusive effect (%) of each NLC-enriched HG was calculated by the following equation:(5)$$Occlusive\;effect\;(\%)=\frac{WVTR_{HG}-WVTR_{NLC\;enriched\;HG}}{WVTR_{HG}}\times 100$$

##### Rheology Study

Rheological measurements of the NLC-enriched HGs were performed using the Kinexus PRO+ rheometer (Malvern^®^, UK) using a 40 mm diameter cone-plate geometry. All the tests were performed at 20 °C. A solvent trap was utilized in all experiments to prevent evaporation and drying.

The characterization of the shear thinning behavior was performed in a steady-state shear flow from 0.01 to 1000 s^−1^ of shear rate. The structure recovery properties were investigated utilizing the rotational approach with alternating (5 cycles) low (0.1 s^−1^) and high (100 s^−1^) amplitude strains with a duration of 100 s each.

The acquired measurements were obtained and treated with rSpace software from Malvern Panalytical^®^ (version 1.76.2398).

##### Evaluation of Antioxidant Effect

The antioxidant effect of NLC1 and NLC2 was evaluated by their ability to scavenge free 2,2′-azino-bis(3-ethylbenzothiazoline-6-sulfonic acid (ABTS^•+^) radicals and prevent the production of thiobarbituric acid reactive substances (TBARS) as a by-product of lipid peroxidation using commercial antioxidant assay kits (Cayman Chemical Company, Ann Arbor, Michigan, USA) in accordance with the manufacturer’s protocol.

In the case of the ABTS assay, the amount of ABTS^•+^ produced was quantified by reading the absorbance at 750 nm in a microplate reader (Synergy H1M2, BioTek, Agilent Technologies, CA, USA), and the antioxidant scavenging activity of each formulation is proportional to the absorbance reduction [42,43,44,45]. The scavenging effect of each formulation was compared to that of Trolox, a water-soluble tocopherol analogue. For this purpose, a calibration curve was plotted from the Trolox standards’ (0 to 0.495 mM) absorbance at 750 nm and the antioxidant concentration (mM) of each formulation was obtained from the linear regression of the standard calibration curve.

In the TBARS assay, malondialdehyde (MDA), a natural product resultant of lipid peroxidation, at high temperature and in acidic conditions, reacts with thiobarbituric acid (TBA) to form an MDA-TBA adduct that was measured colorimetrically at 540 nm [46,47]. For this purpose, a calibration curve was plotted from MDA colorimetric standards’ (0 to 50 μM) absorbance at 540 nm, and the MDA concentration (μM) produced from each formulation peroxidation was obtained from the linear regression of the standard calibration curve.

##### Qualitative Photoprotection Effect

The photoprotection effect of NLC2-enriched HGs was qualitatively assessed using UV Sensitive Paper (Nature Print Paper, USA). Briefly, a small amount of HG was evenly spread at the midpoint of a quartz slide, which was subsequently positioned atop the paper. This reactive paper is impregnated with iron (III) hexacyanferrate (III), Fe[Fe(CN)_6_], also known as Berlin Green, which undergoes a photochemical transformation upon exposure to UV radiation to a water-insoluble chemical iron (II) hexacyanoferrate (II), Fe[Fe_4_(CN)_6_]_3_, known as Prussian Blue. The samples were irradiated with the UV light (UV curing fluorescent lamp with a spectral irradiance of 98.5 W·m^−2^ in the UVA region, i.e., 320 nm to 400 nm) (Techinstro Industries, Nagpur, India) for 3 min, and the paper was then rinsed in water. In the event of UV protection, the water-soluble chemical undergoes a process of dissolution, resulting in the paper’s whitening. Conversely, the water-insoluble complex formed as a result of UV action remains affixed to the paper, turning it blue.

##### Evaluation of the Antimicrobial Effect

The antimicrobial activity of the NLC-enriched hydrogels was tested against three different microorganisms: a fungal species *Candida albicans* (SC 5314, a clinical isolate belonging to the CEB Biofilm Group collection), and two bacteria, one gram-positive, *Staphylococcus aureus* (ATCC 6538, a clinical isolate belonging to the CEB Biofilm Group collection), and one gram-negative, *Escherichia coli* (CECT 434, a clinical isolate belonging to the CEB Biofilm Group collection).

*S. aureus*, *E. coli*, and *C. albicans* were cultivated in a liquid medium by inoculating a single colony in 30 mL of TSB for bacteria and SDB for the yeast. The microorganisms were then incubated for 18 h at 37 °C and 120 rpm. Afterwards, the resultant cell suspension was adjusted to an optical density (OD) below 1.0 at 620 nm for bacteria and properly diluted in TSB to 1 × 10^8^ CFU·mL^−1^. For *C. albicans*, the cell density was further adjusted to 1 × 10^8^ cells·mL^−1^ using a Neubauer hemocytometer.

To the incubation of the microorganisms in the agar, an aliquot of cell suspension (100 μL) was spread on TSA or SDA petri dishes, respectively, for bacteria and yeast. Then, blank disks impregnated with NLC-enriched HGs were placed on top of the agar plate and incubated for 24 h at 37 °C. After the incubation time, the agar plates were left at room temperature for another week. After the incubation period, the bacterial inhibition or reduction halo formed around the samples was photographed to record the results (images captured with the Image Lab™ 6.1. software), and its diameter was also measured with the ImageJ 1.53 t software. All experiments were repeated for at least three independent assays.

#### 2.2.5. Statistical Analysis

Statistical analysis was performed using GraphPad Prism^®^ version 5.0. One-way ANOVA was performed for testing statistical significance in experimental assays. Tukey’s test was applied to explore multiple comparisons. Statistical significance was considered for *p* < 0.05.

## 3. Results and Discussion

### 3.1. Physicochemical Characterization of Nanostructured Lipid Carriers

Since the NLCs developed are intended to be administered topically, their physical properties, such as size and surface charge, should be studied, as these may influence their skin penetration. Moreover, the quality of the formulations produced should be assessed in terms of their colloidal stability under shelf conditions. Figure 2 displays the results for the Z-average size, PDI, and zeta potential values (correlated with the surface charge) of NLC1 and NLC2 produced without placebo and with Q or/and ω_3_.

The physicochemical properties of NLCs’ placebo are not significantly different from those containing Q or/and ω_3_, suggesting that the actives did not affect the original lipid packing of the nanoparticles, and thus the NLCs’ size and PDI are maintained when actives are added (Figure 2A,B). The surface charge of NLCs’ placebo is also not considerably changed when actives are added, indicating that they are incorporated at the core of the NLCs and do not affect the interfacial region.

However, with the exceptions of NLC + ω_3_ (when size is compared) or NLC + Q + ω_3_ (when PDI is compared), the surfactant composition has an impact on the size, PDI, and surface charge of the nanoparticles. Indeed, NLC2 are significantly smaller (*p*-value < 0.01 for NLCs placebo and NLCs + Q + ω_3_; *p*-value < 0.05 for NLCs + Q) but have significantly higher PDI values (*p*-value < 0.05 for NLCs placebo, NLCs + Q, and NLCs + ω_3_) and are more negatively charged than their NLC1 counterparts (*p*-value < 0.001). According to the theory of emulsions, the size reduction may be attributed to the effect of the NLC2 surfactants decreasing the interfacial tension between the lipid and aqueous phases, leading to the formation of smaller droplets and consequently smaller NLCs. Surfactants’ ability to reduce interfacial tension is often attributable to electrostatic and spatial repulsion [48,49], which is inconsistent with the significantly lower (i.e., more negative) zeta-potential values of NLC2. On the other hand, despite NLC2 having smaller sizes than NLC1, their PDI values are significantly higher (*p*-value < 0.05), which can be attributed to Tween^®^ 80, which has been described as increasing the PDI in concentrations greater than 0.5% due to a micellization effect [50]. Regarding the zeta potential values, both NLCs are negatively charged. Even if NLC1 are composed of non-ionic surfactants, the presence of residual free fatty acids in soy lecithin (e.g., phosphatidic acid) [22] or poloxamer (e.g., resultant from polymer hydrolysis) confers a negative net charge to the formulations [51]. NLC2 composition includes an anionic surfactant, DOSS, which justifies the significantly lower (more negative) zeta potential values obtained for NLC2 than NLC1.

To the best of our knowledge, there are very few reported lipid nanosystems developed for simultaneous administration of Q and ω_3_ [24,25,26,27], and only one study prepared NLCs with both components [27]. Moreover, none of these studies intended topical administration. The Z-average sizes of NLC1 + Q + ω_3_ (225 ± 93 nm) and NLC2 + Q + ω_3_ (83 ± 11 nm) were within the size range obtained by Azizi et al. (≈90–315 nm) [27], which also included fish oil and Q encapsulated in NLC, composed of palmitic acid as solid lipid and whey-protein isolate as stabilizer. In agreement with the recommended for lipid-based nanosystems, where a PDI of 0.3 or lower is considered acceptable and indicates a homogeneous population [52], our NLCs’ formulations presented lower PDI values: NLC1 + Q + ω_3_ (0.15 ± 0.08 nm) and NLC2 + Q + ω_3_ (0.246 ± 0.004 nm). Comparatively, Azizi et al. reported larger PDI values characteristic of highly polydisperse samples with multiple particle size populations (≈0.35–0.60) [27]. Other reported Q-loaded NLCs using linseed oil (instead of ω_3_) obtained a size range of 89.2 ± 0.2 to 95.6 ± 0.3 nm [53], similar to NLC2 + Q + ω_3_ (83 ± 11 nm), but the authors used a much higher surfactant concentration (12% versus 3% *w/w*) in the former formulations, which is against the research trends due to possible detrimental effects (e.g., production cost and environmental discharge) of high surfactant ratios [25,26,54].

Colloidal stability is very important when the loading cargos, like Q and ω_3_, are prone to oxidation, since it helps to mitigate the interactions with reactive oxygen species (ROS) present at the interface or surrounding aqueous phase. The colloidal stability of the NLCs under storage conditions was evaluated to compare the physicochemical properties from the production day, T1, until 4 weeks, T4 (Figure 2), and no significant changes were observed for both NLC formulations. Additionally, the macroscopic appearance shows no phase separation (Appendix A). These observations imply that the NLCs developed kept their colloidal stability and surpassed the previously published experiments in this regard. Azizi and co-workers developed NLCs containing both bioactives that show thermodynamic stability, but they were only tested over a short period of time (4 days) [27]. Moreover, other reported lipid nanosystems containing Q and ω_3_, such as liposomes, have shown weaker colloidal stability since their size increased by more than 200% in 24 h, and the authors also observed a PDI increase from 0.300 to 0.750 during the storage period (24 days) [24].

The EE of Q in NLC1 and NLC2 (with and without ω_3_) was assessed to be higher than 99.0% (<LoD = 0.002 µg·mL^−1^ based on the calibration curve and the residual standard deviation results). Considering Q maximum aqueous solubility (2.150 µg·mL^−1^ at 25 °C) [55], NLCs were able to encapsulate 500 µg·mL^−1^ of Q, outperforming Q dispersed in an aqueous medium by 233 times. The DLs of Q in NLC1 and NLC2 (with and without ω_3_) were respectively 0.62% and 0.43%. Comparing the DLs of Q with those reported for lipid nanosystems containing Q and ω_3_, the values obtained are within the same range (0.27–0.76%) [24,25]. Other Q-loaded NLCs (without ω_3_) reported much smaller DL values (0.094–0.280%) [53].

During the development of NLCs, DSC studies are often employed in order to assess the thermal behavior of the lipid phase. One of the prerequisites for their production is that the mixture of solid and liquid lipids must be solid at body and room temperature.

DSC thermograms of the oils (ω_3_ and Labrasol^®^); solid lipids (Precirol ATO^®^ 5 and Gelucire^®^ 50/13); and Q (raw or in the lipid melted mixture) were analyzed (Figure 3).

The thermograms of ω_3_ and Labrasol^®^ did not show any thermal events since these components are oils at room temperature and do not degrade below 145 °C.

The melting peak of solid lipid Precirol^®^ ATO5 occurs at 56.9 °C and displays a slight shoulder of decreased enthalpy at roughly 52.0 °C (Appendix A). These two melting events have been well documented in the literature and are caused by the presence of different polymorphic forms found in complex glyceride combinations such as Precirol^®^ ATO5: metastable α-polymorph (shoulder) and stable β-form (peak) [56]. The other solid lipid, Gelucire^®^ 50/13, is also a mixture of crystalline and amorphous glycerides as shown by the multiple endothermic events of the thermogram [57]. The raw Q thermogram presents an endothermic peak at 129.2 °C, possibly related to its reported thermochemical transition involving structural relaxation and decomposition [58]. Identical endothermic events of each individual solid lipid component may be identified in the thermogram of Q in the lipid melted mixture (that simulates the NLCs’ lipid phase), but the endothermic peaks are broader and have lower associated enthalpies (Appendix A). These changes are mostly due to interactions between the solid lipids and oils during the melting process, which result in a less ordered lattice that is beneficial for encapsulating larger amounts of actives [59,60]. The absence of the Q endothermic peak in the lipid-melted mixture thermogram (Figure 3) suggests a possible dissolution of Q in the NLCs’ lipid phase.

To conclude, both NLCs have good Q loading capacity, greatly improving their solubility, as evidenced by the melted lipid phase thermogram, which shows no Q thermal events. Furthermore, both Q-loaded NLCs are solid at room and body temperatures, which is desirable for thermal stability and longer-term active release. However, the dimensions of NLC2 are more appropriate for topical administration, as they fall within the optimal size range of 100 to 200 nm [3,61]. In addition, the significance of small sizes lies in their relevance to topical applications, as the ability of drug delivery systems to penetrate the *stratum corneum* diminishes with an increase in size [3,61]. Indeed, lipid nanosystems with a size range of 20–200 nm are described as having the potential to accumulate on skin irregularities such as pores and follicles, thereby facilitating closer interaction between actives and the skin [62]. Nevertheless, the size constitutes a constraining element for rigid drug delivery systems, given their inherent structural inflexibility. According to the literature, the presence of components that disturb lipid packing (e.g., ω_3_, which is also known as “edge activators”) can confer greater flexibility and malleability, thereby enhancing their ability to permeate the skin with greater efficacy [23]. NLC2 may have longer shelf stability, as it has been pointed out that, among other factors contributing to nanoparticles’ colloidal stability, zeta potential values equal to or greater than |±30| mV display electrostatic repulsion and a lower propensity to aggregate [63].

### 3.2. Potential Pharmaceutical and Therapeutic Performance

Criteria can be utilized to predict the pharmaceutical and therapeutic potential of topical cutaneous formulations in vitro. The first criterion is to study the release profile of the actives in a medium mimicking skin conditions, i.e., pH 5.6 and temperature of 32 ± 3 °C [35,36]. The release profile of Q from NLC1 and NLC2-enriched HGs is presented in Figure 4A.

The nonlinear fitting of Q release with several mathematical models (first-order, Korsmeyer-Peppas, Weibull, Higuchi, and Gallagher-Corrigan) was tested and presented in Appendix A. The Weibull model was the one that resulted in higher adjusted R^2^ values (R^2^ = 0.999 for the NLC1 + Q, NLC2 + Q, and NLC2 + Q + ω_3_; R^2^ = 0.994 for the NLC2 + Q + ω_3_ formulation). The values of the b parameter superior to 1 are indicative of a complex release mechanism: the rate of release initially increases non-linearly up to the inflection point and thereafter decreases asymptotically [64]. NLCs released Q in a sustained manner, reaching a maximum of 53.8 ± 0.9% over 24 h from NLC1 + Q + ω_3_, with a kinetic constant of *k* = 1.4 × 10^−3^ ± 1 × 10^−4^ min^−1^ (Appendix A, first-order fitting model, Appendix A), which was not significantly different from the Q release rate from NLC1 + Q. The release rates of Q from NLC2 were significantly lower than those obtained for NLC1 (*p*-value < 0.001, for paired observations NLC1 + Q vs. NLC2 + Q; and *p*-value < 0.01, for paired observations NLC1 + Q + ω_3_ vs. NLC2 + Q + ω_3_), with cumulative releases at 24 h of 38.4 ± 0.4% and 41.9 ± 0.5% for NLC2 + Q and NLC2 + Q + ω_3_, respectively. The key variables influencing the release profiles from lipid nanosystems are particle size, type of surfactant, and actives’ concentration [53]. The Q loading content in the two types of NLCs was similar, and thus only the particle size and surfactant composition can have a role in the distinct release profile. Although NLC2 particles have smaller surface areas and thus provide higher interactions with the release aqueous media, their Q release is more sustained. However, as with other types of particles, the interfacial tension effect can affect not only particle size but also drug distribution within the particle, ultimately controlling the duration of release [65]. Thus, the different surfactant composition of NLC2, which, as previously stated, decreases the interfacial tension between the lipid and aqueous phases, may provide a different distribution of Q within the lipid core during the emulsification process, reducing Q’s accessibility to the solvent media and thereby slowing its release.

Although Huang et al. did not present the kinetic release parameters, a maximum Q of ≈30–45% was released from their reported NLCs [53], whereas a maximum Q of ≈38–54% was released from the NLCs herein presented. The release profile obtained in our study was also more sustained since only ≈4–7% of Q was released within 120 min compared to the reported Q release (≈12–25%) [53]. The sustained release of Q is advantageous in terms of ensuring the chemical stability of this bioactive, and in this regard, NLCs provide a protective carrier to slow down Q aqueous degradation. In fact, we found that Q-loaded NLC-enriched HGs released around 20% less Q one month after preparation (i.e., ca. 20% Q suffered hydrolysis); however, if Q was left free in the HG (non-encapsulated), the reduction in Q release is 30% less after only one day, suggesting ca. 30% hydrolysis in just one day when Q was not protected inside the NLCs (Appendix A).

The in vitro diffusion across a mimetic *stratum corneum* lipid membrane (Figure 4B) revealed distinct Q permeation profiles from the developed NLC-enriched HGs, with significantly different Q maximum diffusion flux (*Jss*) values (*p*-value < 0.001). Our NLC-enriched HGs and promoted Q permeation in the following order: NLC2 + Q + ω_3_ (*Jss =* 2.2180 μg·cm^−2^·h^−1^) > NLC2 + Q (*Jss =* 1.9720 μg·cm^−2^·h^−1^) > NLC1 + Q + ω_3_ (*Jss =* 0.7906 μg·cm^−2^·h^−1^) > NLC1 + Q (*Jss =* 0.2657 μg·cm^−2^·h^−1^). These findings support the generally accepted idea that spherical particles with diameters smaller than 100 nm possess higher permeability and demonstrates the role of ω_3_ as an “edge activator” when present in the composition of NLCs, probably due to its ability to disrupt the lipid packing, conferring deformability to the lipid matrix and thus increasing permeability [23]. *Jss* calculation enables chemicals to be classified according to their inherent maximum skin permeation, as proposed by Kroes et al. [66,67]. Applying this classification to the *Jss* values obtained indicates that NLC1 (with or without ω_3_) confers to Q a medium-low permeation capacity, whereas NLC2 (with or without ω_3_) confers to Q a medium-high permeation capacity.

Although the permeation results are encouraging in terms of developing a topical formulation for Q administration, this study has a limitation that must be considered when interpreting its findings. A direct comparison of the Q-loaded NLC-enriched HGs with free Q in solution or free Q in the HG was not possible due to methodological issues. Indeed, Q is barely soluble in water [53], and a solution or dispersion of Q in the HG would require an ethanolic content, which would cause solubilization of the *stratum corneum* lipid model used in the permeation assay. In any case, this is not an exclusive problem of the model used, because in a real skin topical application, ethanol, frequently applied as a permeation enhancer, works as such due to its ability to solubilize the *stratum corneum* lipid matrix, which is not a desirable situation as it affects the skin barrier function against several external aggressions. Despite this methodological constraint, we can compare the Q permeation achieved in NLC-enriched HGs to the free Q permeation predicted in silico. According to the algorithm developed by Potts and Guy [68], the theoretical *Jss* of free (i.e., non-encapsulated) Q through human skin was calculated to be ≈ 0.003480 μg·cm^−2^·h^−1^, which is consistent with a predicted low permeation capacity. Consequently, the *Jss* of the developed Q-loaded NLC-enriched HGs is higher than the *Jss* of free Q, as evaluated in silico. Hence, the developed NLCs are expected to act as permeation enhancers because they significantly increase the Jss values of Q compared to what was expected for this compound when administered freely in the skin.

In addition to improving permeability, it is critical to determine whether the formulations may limit water loss from the skin, i.e., have an occlusive effect that can interfere with human skin’s natural transepidermal water loss [69,70]. The occlusive effect has the potential to enhance the hydration of the *stratum corneum*, thereby affecting the percutaneous absorption and efficacy of bioactives [71]. In this study, the occlusive effects of NLC-enriched HGs were compared, as shown in Figure 4C. All NLC-enriched HGs presented a high occlusive effect, ranging from a 13 ± 5% (NLC1 Placebo) to a 34 ± 1% (NLC1 + Q) decrease in WVTR (Figure 4C). The same effect has been reported for lipid nanoparticles produced for topical applications, even when no active compounds are present [23,72], which can be attributed to the lipid film layer’s adhesion on the skin’s surface, which aids in the occlusive effect and skin hydration. Han et al. reported Q-loaded SLNs with a higher occlusion effect (≈30–60%) [72], but several factors can influence the occlusion, such as particle size, lipid concentration, and crystallinity [73]. Moreover, if similar sizes and lipid contents are used, SLNs generally have a higher occlusion factor compared to NLCs due to the higher crystal structure of the former lipid nanoparticles.

Pharmaceutical and cosmetic formulations’ flow parameters have a significant impact on their application and acceptance [74]. Understanding their rheological properties is therefore critical for improving processing effectiveness and producing finished products that are acceptable to consumers in terms of spreading and adhering to the skin as well as removal from a tube or jar. The rheological analysis of NLC-enriched HGs is presented in Appendix A and Figure 4D. All the HGs exhibited a typical flow curve of non-Newtonian pseudoplastic behavior, which was visible from the viscosity decrease with the increasing shear rate (Appendix A). This has already been reported for other lipid nanosystem-enriched HGs [23,29,75]. Additionally, NLC1 is more viscous than NLC2. No hysteresis was detected in any of the formulations, at least in the period of analyses, i.e., the viscosity of the formulations did not change since the ascending and descending curves of the rheograms are practically overlapped, indicating absence of thixotropy [76]. In terms of topical cutaneous applications, this indicates that the pseudoplastic behavior of HGs promotes their spreading, but they can quickly regain their consistency and elastic characteristics after the shear stress is removed, favoring their retention at the skin surface [76,77]. A fatigue test was also performed to confirm the HGs’ structural recovery (Figure 4D). A rotational technique was used to apply five cycles of lower and higher shear rates (0.01 s^−1^ and 100 s^−1^) for 100 s each. A substantial drop in viscosity was found at increasing shear rates. However, at cessation of the higher shear rates, the viscosity was recovered. This quick recovery of viscosity, combined with the HGs’ shear thinning behavior, permits a good spreading of the product in the skin when a massage is applied and also allows a good extrusion via a tube or jar, which is important for cosmetic applications. Figure 4D also confirms that NLC1-enriched HGs have similar viscosity to bare HGs and higher viscosity than NLC2-enriched HGs; however, despite being less viscous, these later formulations are still able to regain their consistency and elasticity after shear stress is removed, indicating that they are also adequate for skin retention after spreading.

Among the deleterious environmental elements (e.g., sun light, pollution, and microorganisms), daily exposure to UV rays consistently produces ROS, which speed up the aging process and cause photo-oxidative damage to skin [2,4,5]. Moreover, ROS that can be generated by lipid peroxidation processes in lipid-based nanosystems, either during production steps or during storage, should be assessed in terms of formulation quality and colloidal stability. In this regard, we evaluated two types of antioxidant activity: (i) the scavenging effect of free radicals by the ABTS assay (Figure 5A) and (ii) the protective effect against peroxyl radicals evaluated by MDA concentration (Figure 5B), which is a common indicator of the degree of peroxidation.

The radical scavenging effect of Q is well documented [78,79] and herein demonstrated to be superior to ω_3_ (Figure 5A), which can be explained by the presence of multiple hydroxyl groups in the Q structure [53,78,80], whereas ω_3_ contains only one of these scavenging moieties as part of its carboxyl group (Figure 1). Despite both Q and ω_3_ demonstrating individual ABTS^•+^ scavenging effects, the total antioxidant activity of their mixture (Q + ω_3_) was not a simple addition of each free component effect, as it depends on their interaction, which can be synergetic, antagonistic, or additive [53]. The scavenging effect of ABTS^•+^ radical was the highest for Q-loaded NLC2, with or without ω_3._ This antioxidant scavenging effect was significantly superior (*p*-value < 0.001, Appendix A) to their free counterparts, most likely due to the chemical stability conferred by NLC loading (Appendix A), leading to Q preserved antioxidant activity when loaded in lipid nanoparticles, as also demonstrated by other authors [81]. Q-loaded NLC1, with or without ω_3_, also demonstrated a scavenging effect of the ABTS^•+^ radical; however, it was significantly lower than NLC2, which can be due to the different radical diffusion according to the distinct rheological NLC properties. Indeed, as documented by Azizi et al., highly viscous lipid nanosystem dispersions may decrease the diffusion of free radicals [27]. As a result, the ABTS^•+^ radicals produced in aqueous media will have a harder time diffusing and meeting Q-loaded components in more viscous NLC1, which justifies the smaller scavenging effect obtained in NLC1 + Q when compared to free Q. The inclusion of ω_3_ in Q-loaded NLC1 favors the fluidity of the formulation (i.e., reduces the viscosity of the formulation, as seen in Appendix A), increasing the accessibility of Q to the ABTS^•+^ radicals, and thus NLC1 + Q + ω_3_ has a similar effect to the free form of Q and Q + ω_3_.

Because MDA is a byproduct of lipid peroxidation, higher [MDA] indicates higher lipid peroxidation (Figure 5B). According to the literature, 7 to 8 mg MDA/kg is an acceptable threshold for determining whether encapsulated fish oil provides adequate oxidation protection [82]. Based on this threshold, our results in Figure 5B show that all NLCs (MDA between 0 and 6 mg/Kg. Appendix A) can be stored for at least 4 weeks, demonstrating suitable oxidation stability and corroborating colloidal stability. Free Q, as expected, did not produce MDA because it is not prone to the lipid peroxidation process, which affects the free fatty acid chains of ω_3_. Interestingly, the mixture of these two actives in the free form greatly potentiates the MDA production, possibly due to the fact that Q can undergo autoxidation under thermal heating conditions [83,84] similar to those used to promote peroxidation, forming several reactive adducts that may propagate lipid peroxidation of ω_3_. It is also worth noting that the reported protective studies of Q against lipid peroxidation of fish oil involve short-term assays, revealing a significant loss of Q protection within 5 days (a much shorter time than the 4 weeks herein evaluated) [85]. The MDA production was the lowest for Q- and/or ω_3_-loaded NLC2. This effect indicates high protection against lipid peroxidation and was significantly superior to NLC2 placebo (*p*-value < 0.001, Appendix A), most likely because placebo formulations are composed of lipids, which are prone to peroxidation in the absence of protective antioxidant agents. The same trend can be observed for NLC1; however, Q- and/or ω_3_-loaded NLC2 produce significantly lower [MDA] (Appendix A), possibly due to NLC2′s lower viscosity, which promotes a more homogenous distribution of the protective antioxidant agents in the NLC and a better diffusion of peroxyl radical towards these protective agents. A similar observation was reported by Azizi et al. [27].

Given the significantly higher antioxidant effect demonstrated by NLC2, HGs enriched with these formulations were used for subsequent photoprotection and antibacterial tests. The photoprotection was qualitatively evaluated using UV-sensitive paper (Figure 6A).

The UV-sensitive paper without any applied HG or with application of HG without NLCs were used as negative controls. The blue color of the paper indicates its inability to be photoprotective since the water-insoluble chemicals remained fixed after washing, whereas the white colorations are indicative of photoprotection. As shown in Figure 6A, even the NLC2 placebo-enriched HG showed some photoprotection compared to negative controls, which is in agreement with the light scattering physical effect that the lipid nanoparticles possess to block UV radiation [22]. As expected, the incorporation of the active (Q and/or ω_3_) boosted overall photoprotection, as seen by the whiter color compared to NLC2 placebo-enriched HG. This photoprotection effect is consistent with studies demonstrating that Q or ω_3_ are promising strategies to ameliorate skin photodamage [22,86,87,88,89].

Given the NLC-enriched HGs’ antioxidant properties, an additional crucial quality parameter of these formulations would be their antimicrobial effect. In fact, formulations that combine antioxidant and antimicrobial ingredients may diversify the skin’s commensal microbiome, improving the health of the epithelium. In this regard, in order to verify if NLC-enriched HGs exhibit some type of antimicrobial activity, tests were carried out using the disk agar diffusion method against three microorganisms (*S. aureus*, *E. coli*, or *C. albicans*). No inhibitory effects were observed in *E. coli* or *C. albicans* growth (Appendix A). The results for *S. aureus* are shown in Figure 6B.

Free Q and Q + ω_3_ show an inhibition zone (transparent zone, free of bacteria), with an inhibition diameter of 11.3 ± 0.2 mm and 10.1 ± 0.2 mm, respectively, around the samples’ discs where no bacterial growth occurs (Figure 6B**,** first row). The inhibition of *S. aureus* growth presented by free Q + ω_3_ is most probably due to the Q antibacterial effect since free ω_3_ does not present an inhibitory halo and only one selvedge is observed. In fact, according to Wang et al. [90], gram-positive bacteria are more sensitive to the bacteriostatic activity of Q. This flavonoid damages the cell wall and membrane, compromising its permeability and inhibiting protein synthesis. Within the NLC-enriched HGs, only NLC2s have shown some reduction in *S. aureus* growth (Figure 6B**,** third row).

HGs enriched with NLC2 placebo, NLC2 + Q, and NLC2 + Q + ω_3_ all show a reduction in bacterial growth as demonstrated by the halo surrounding their sample disks, with diameters of 16.1 ± 0.2 mm, 17.4 ± 0.6 mm, and 18.8 ± 0.5 mm, respectively. Despite the lack of any active ingredient responsible for antibacterial activity, NLC2 placebo-enriched HG nevertheless reduced bacterial growth, which was most likely due to the NLC2s’ surfactant composition (Tween^®^ 80 and DOSS). In fact, when a disk impregnated with the two surfactants was placed on the agar plates, *S. aureus* growth was reduced (Appendix A). The inhibitory effect of Tween^®^ 80 and/or other surfactants against *S. aureus* has already been reported [91]. However, the surfactant mixture does not cause any growth reduction in *E. coli* or *C. albicans* (Appendix A).

The results for HGs enriched with NLC2 + Q or NLC2 + Q + ω_3_ are easily explained by the Q activity against *S. aureus* [92,93], but the effect of Q is diminished in the formulation as there is only a reduction in *S. aureus* growth. The antibacterial effect is appealing for skincare and protection as it may increase microbiome diversity, which is thought to be a preventive health factor. In this context, the observed bacteriostatic effect may be more beneficial in ensuring that skin flora is not compromised than a bactericidal effect, which could promote skin microbiome imbalance and opportunistic pathogen overgrowth.

The formulations were tested against *S. aureus*, *E. coli*, and *C. albicans* as representatives of microorganisms belonging to the groups of gram-positive bacteria, gram-negative bacteria, and yeasts, respectively. Future studies could include other bacteria belonging to the skin microbiome as well as antibiotic-resistant strains such as MRSA. However, although not within the scope of the current study, it would be interesting to continue research into the antimicrobial activity of these formulations to study microbial interactions in co-culture, identify interaction patterns, and characterize a microbial strain’s inhibitory potential of the current formulations against human and environmental pathogens. Co-culture studies would also be important to confirm the non-harmful effect on skin flora and the further potential to increase the skin microbiome. Moreover, it would be interesting to perform additional research on the interdependence of skin microbiome and Q-loaded NLC-enriched HGs. Indeed, as extensively reported, Q and other flavonoids can modulate the intestinal microbiota, providing significant health benefits and preventing chronic diseases [6,94]. The intestinal microbiota, on the other hand, affects flavonoids bioactivity and bioavailability by hydrolyzing them to catabolites that further enhance their role in the microbiome and health promotion [6,94]. Throughout history, topical application of the same flavonoids used in dietary supplements has resulted in the development of healthy and glowing skin. Therefore, it is reasonable to speculate that the skin microbiome and flavonoids, such as Q, may interact in a manner similar to that observed between flavonoids and intestinal microbiota, and this deserves further investigation.

In summary, as observed from the physicochemical characterization, NLC2 has shown better pharmaceutical and therapeutic performance. In fact, NLC2 displayed a more sustained Q release, indicative of higher preservation of the bioactive, while favoring bioactive permeation when in contact with a *stratum corneum* mimicking membrane. Additionally, a positive effect on skin permeation is achieved due to NLC-enriched HGs’ occlusive effect.

In terms of the antioxidant impact, Q-loaded NLC2, with or without ω_3_, was the most effective in scavenging free radicals and protecting against lipid peroxidation. In addition, NLC2-enriched HGs also exhibited photoprotective and antibacterial effects against *S. aureus*. Taking into account the set of results of this study, NLC2-enriched HG proved to be promising for topical administration to prevent skin aging and other cutaneous damaging processes.

## Figures and Tables

**Figure 1 pharmaceutics-15-02078-f001:**
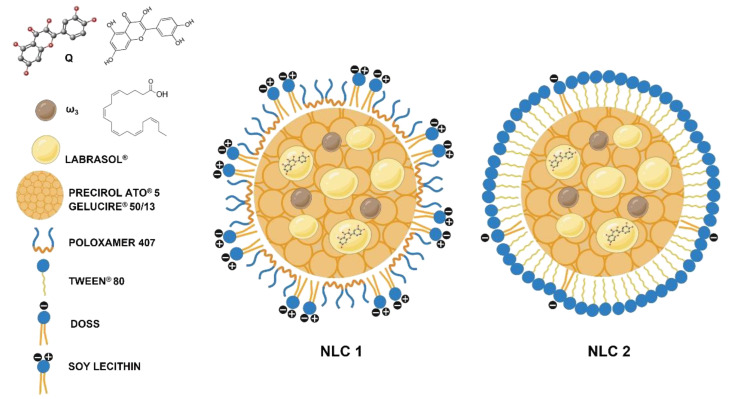
Schematic representation of nanostructured lipid carriers (NLCs) with or without quercetin (Q) and/or omega-3 fatty acid (ω_3_). Created with BioRender.com.

**Figure 2 pharmaceutics-15-02078-f002:**
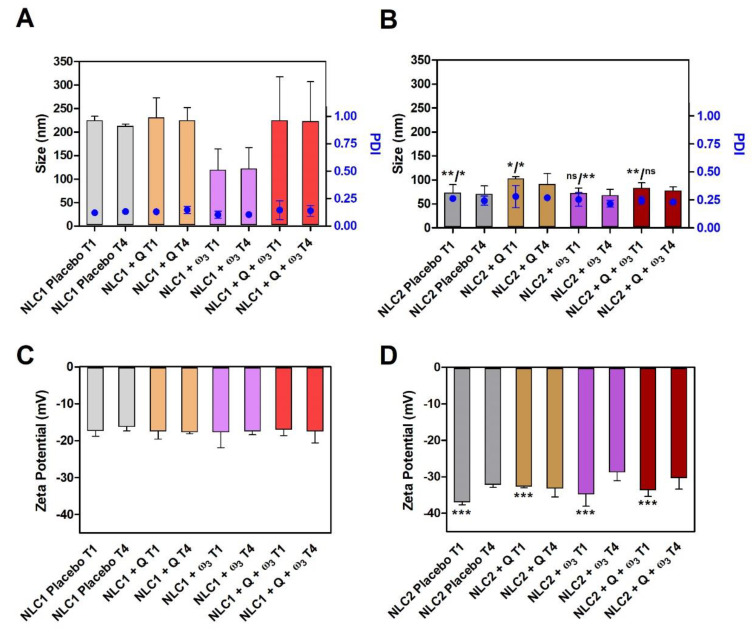
Stability characterization of NLC1 (**A**,**C**) and NLC2 (**B**,**D**) during storage at 4 °C. The data represent the mean ± SD of three independent measurements, each from three different formulation batches. Measurements of size (bars), PDI (dots) (**A**,**B**), and surface charge (Zeta-Potential (mV)) (**C**,**D**) are from day of preparation (T1) and after 4 weeks (T4). Comparisons were performed using one-way ANOVA test and Tukey’s multiple comparison test for the following paired observations (size/PDI in (**B)** and Zeta potential in (**D**)): NLC1 Placebo T1 vs. NLC2 Placebo T1; NLC1 + Q T1 vs. NLC2 + Q T1; NLC1 + ω_3_ vs. NLC2 + ω_3_ T1; NLC1 + Q + ω_3_ T1 vs. NLC2 + Q + ω_3_ T1. ns—no statistical significance; *** *p*-value < 0.001; ** *p*-value < 0.01; * *p*-value < 0.05. No statistical significance was found when T1 and T4 were compared within each NLC formulation; nonetheless, in this case, for clarity, ns was not displayed in the figure.

**Figure 3 pharmaceutics-15-02078-f003:**
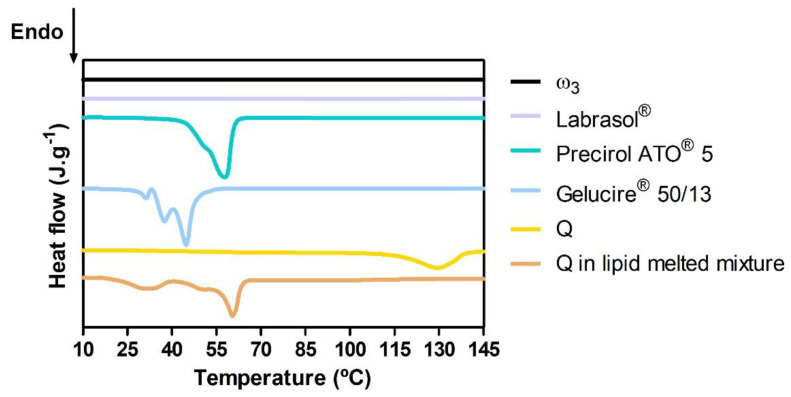
DSC thermograms of the NLC components and Q in the lipid melted phase. After 145 °C, no thermal events were observed.

**Figure 4 pharmaceutics-15-02078-f004:**
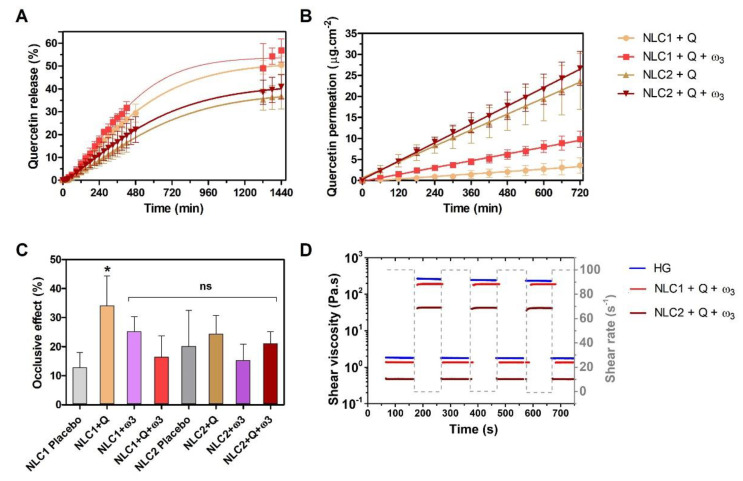
Evaluation of pharmaceutical performance of NLC-enriched HGs: release and permeation profile of Q (**A**,**B**); occlusive effect (**C**); rheological behaviour (**D**). The data represent the mean ± SD. Comparisons of occlusive effect (**C**) were performed using one-way ANOVA test and Tukey’s multiple comparison test, obtaining statistical significance for the following paired observations: NLC1 Placebo vs NLC1 + Q; * *p*-value < 0.05. Other comparisons with NLC1 Placebo have no statistical significance; ns.

**Figure 5 pharmaceutics-15-02078-f005:**
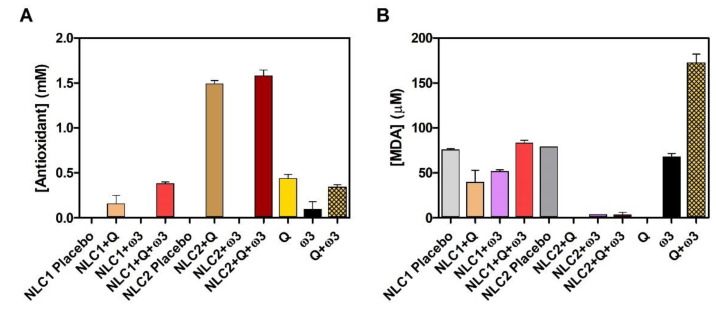
Evaluation of antioxidant effect of NLCs. Scavenging effect of radical ABTS^•+^ expressed as antioxidant concentration (mM) and measured 3 days after NLCs production (**A**); Lipid peroxyl radical LOO^•^ production expressed in terms of MDA concentration (μM) and measured 4 weeks after NLCs production (**B**) The data represent the mean ± SD. Comparisons of [Antioxidant] (mM) (**A**) or [MDA] (μM) (**B**) were performed using one-way ANOVA test and Tukey’s multiple comparison test and are presented in Appendix A.

**Figure 6 pharmaceutics-15-02078-f006:**
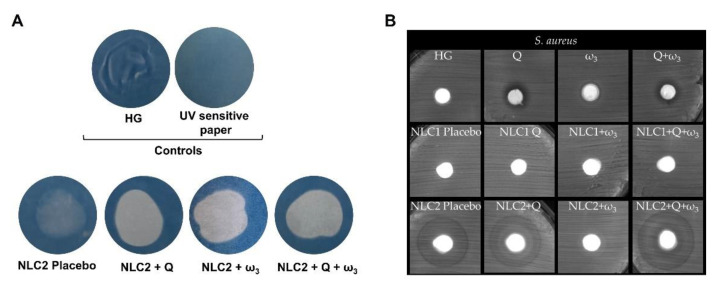
Evaluation of photoprotective effect (**A**) and antibacterial activity against *S. aureus* (**B**) of NLC-enriched HGs.

## Data Availability

Not applicable.

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
