# Peer review of "Nanostructured Lipid Carriers Enriched Hydrogels for Skin Topical Administration of Quercetin and Omega-3 Fatty Acid"

_pharmaceutics, 2023, doi:10.3390/pharmaceutics15082078_

Round 1
Reviewer 1 Report
Thank you for the opportunity to review the manuscript by Lucio and colleagues. The goal of these studies is to test the ability of a novel hydrogel to deliver the antioxidant quercetin and omega-3 fatty acids. The topic has value as there is a need for vehicles for relatively unstable agents for skin application.
The methodologies are sound. The studies are relatively exhaustive. The modified NLCs allowed more sustained release and photoprotective effects. Finally, these agents blocked Staphylococcus aureus but not Candida/E. Coli growth. Overall this is a very important and well-written manuscript, and the outcomes are important for topical application of relatively unstable agents.
I have a few questions that are very minor in nature.
1. The antibacterial effects of the NLCs selectively for Staph—this is interesting yet the authors do not provide any explanation. One fear is that these formulations could inhibit “normal” flora such as S. epi and others, and allow gram negative (eg E. Coli) and/or fungal (eg, Candida) overgrowth. What further testing do the authors feel is warranted as this would be an important issue in bringing these formulations to market.
2. Could the authors provide more detail as the source of the membrane used in the Franz diffusion chambers?
3. Could the authors provide more detail as the UV source used (model and manufacturer)?
Reviewer 2 Report
The work presented by Lúcio and colleagues focuses on the development of quercetin and omega-3 fatty acids-encapsulated nanostructured lipid carriers for topical administration. The authors characterized the formulation and evaluated its performance in terms of release, permeation, rheological behavior, UV protection and antibacterial/antifungal activity. Overall, the manuscript is well written and presents interesting results.
Some aspects, particularly those related to quercetin release and permeability, require further clarification before being considered for publication in Pharmaceutics.
1. Excipient selection highly depends on the characteristics of the drug(s) to be encapsulated. Lipid selection should be supported by solubility studies (at least in precirol, labrasol and gelucire (https://doi.org/10.3390/biom12020223).
2. The relationship between the validated HPLC method and the one used by the authors is not well established. The sentence is misleading. The authors should avoid mentioning the “validation” of the method, as they have not fully/partially validated it and simply state “determined by a high performance liquid chromatography (…)”
3. Is there any impact on the colloidal properties of the NLCs upon the dispersion and particularly, pH correction?
4. Why did the authors use a 3D assay for characterizing the quercetin release? Franz diffusion cells are a far better technique to describe drug release of topical products.
5. Is the stratum corneum lipid membrane model validated elsewhere? A reference should be added to support the method.
6. The relationship between surfactants and interfacial tension is not well described. According to the theory of emulsions, the reduction of the interfacial tension between the lipid and aqueous phases leads to the formation of smaller droplets and consequently smaller NLCs. This should be critically revised throughout the manuscript.
7. The association between particle size and Q release (smaller NLCs – increased surface area but lower Q release) is not well documented (Figure 4A).
8. Why does Q release decreases after NLC storage? This phenomenon should be addressed in more detail (Figure S2). SD ranges are missing from the graph (red dots and stars).
9. The performance of the NLCs (in terms of release but particularly, permeation) is not well established, as the authors did not compare it to a Q solution. How can we be sure that NLCs are better than a solution or Q hydrogel? The authors have used a synthetic stratum corneum model and have compared the experimental NLC Jss values with those obtained from the literature for completely different models/experimental conditions.
Minor:
Figure 2. The PDI axis should be corrected, as PDI is ≤ 1.
Figure 3. DSC analysis was conducted up to 200 °C, but the graph only displays data until 145 °C.
L 449 – 452 The reference to the ± 30 mV is flawed, at least in what concerns SLNs/NLCs, as there are other factors impacting stability. Several SLN/NLCs show great stability with ZP < ± 30 mV and vice versa. This sentence should be removed.
Minor english spelling errors.
Reviewer 3 Report
Dear authors,
A negative point is represented about no data related to skin bioavailability in the discussion. The microbiota bioactivity and bioavailability of these functional compounds are important details that should be mentioned and these details could be part of the future paper valorization.
Best regards!
Reviewer 4 Report
The article “Nanostructured lipid carriers enriched hydrogels for skin topical administration of quercetin and omega-3 fatty acid” by Marlene Lúcio and colleagues reports the preparation of two types of lipid formulations for the delivery of quercetin, with or without o3 co-delivery). The article is well-presented and organized, nevertheless some points need to be addressed to justify its significance.
Most importantly, no cellular experiments have been reported at all. A simple cytotoxic study (e.g. MTT) must be performed (at least) to evaluate the biocompatibility of the prepared formulations at the concentrations tested in the already-reported experiments.
The DSC thermographs of all the complete systems (e.g. NLC1 and NLC2 alone, with Q, with o3, and with both Q+o3) must be included in addition to those of the precursors (which serve as controls).
Please convert rpm to either g or rcf.
Use the same precision for average numbers and corresponding standard deviations (e.g. lines 656-663).
Some English and (mostly) editing errors are present.
Round 2
Reviewer 2 Report
The authors have addressed my concerns and suggestions. I have no further comments.
Author Response
We value the reviewer's feedback and willingness to revise our manuscript.
Reviewer 4 Report
The revised version of the article did not address the major issues present in the previous version. In my opinion, they are necessary to improve the quality and the impact of the article as it already lacks novelty.
Despite the authors claim that the components are considered by FDA as "generally" safe (and we agree on that), the cellular experiments at the concentrations test in the article (MTT, LDH, etc) are necessary to validate that hypothesis (otherwise almost nobody would perform those experiments).
Similarly, TGA analysis MUST be performed on the final system and not just on the component. Plenty of articles in the literature report those analyses on similar nanoparticles and water is not an issue but it helps to understand the general behaviour of the complex.
Author Response
We appreciate the reviewer's comments and willingness to revise our manuscript. We performed DSC analysis of the final systems NLC1 and NLC2 as suggested by the reviewer, and we added a new figure comparing the bulk materials DSC (physical mixture) and final systems (NLC 1 and NLC2). Table S3 now includes thermogram details of the components of the formulations and of the final formulations. As expected, the lipid phase contributes significantly to the thermal events of the final formulations; thus, thermograms of the final systems are qualitatively similar to those of the bulk materials, confirming that the final systems are solid at room and body temperature.